# Resilience of Medication Adherence Practices in Response to Life Changes: Learning from Qualitative Data Obtained during the COVID-19 Pandemic

**DOI:** 10.3390/healthcare9081048

**Published:** 2021-08-13

**Authors:** Mushfique Ahmed, Carly Wheeler, Bryony Dean Franklin, Rabia Begum, Sara Garfield

**Affiliations:** 1UCL School of Pharmacy, 29-39 Brunswick Square, Bloomsbury, London WC1N 1AX, UK; mushfique.ahmed.16@ucl.ac.uk (M.A.); bryony.deanfranklin@ucl.ac.uk (B.D.F.); rabia.begum.17@ucl.ac.uk (R.B.); 2Imperial College Healthcare NHS Trust, Fulham Palace Rd, Hammersmith, London W6 8RF, UK; carly.wheeler1@nhs.net; 3Imperial NIHR Patient Safety Translational Research Centre, St Mary’s Hospital, Praed St., Paddington, London W2 1NY, UK

**Keywords:** adherence, medication, resilience, housebound

## Abstract

Nonadherence to medicines is widespread and can adversely affect health outcomes. Previous research has identified that patients develop their own strategies to assist with adherence. However, such research has not focused on how the helpfulness of these strategies may change in response to changes in patients’ circumstances. This study aimed to explore resilience of medication adherence to life changes. It involved secondary thematic analysis of the verbatim transcripts of 50 semi-structured interviews that were conducted with adults who were advised to shield or were over the age of 70 during the first wave of the COVID-19 pandemic in the UK. Interview data suggested that resilience of medication adherence varied between participants. Participants either reported that they had not used any specific strategies to remind them to take their medicines prior to the pandemic, that the strategies that they had employed prior to the pandemic remained effective during the pandemic, that they had needed to make some adjustments to the strategies used, or that the strategies they had used were no longer effective. In addition, beliefs about medicines and motivation to take them were altered for some participants. These findings suggest that challenges associated with medication adherence do not always remain stable over time and that healthcare professionals need to continue to monitor and support medication adherence long-term.

## 1. Introduction

Medication is one of the most common forms of treatment, with almost 70% of the UK population having received prescriptions for one or more medicines in their lifetime [1]. However, it is estimated that 30–50% of all medicines prescribed worldwide for long-term conditions are not taken as recommended by healthcare professionals [2]. Nonadherence to medication can reduce its effectiveness, lead to worse health outcomes, and increase use of healthcare resources [3]. Nonadherence can be classed as unintentional, where execution of the agreed action is intended but does not occur, or intentional, where planned changes to the agreed action take place [4].

A large number of studies have explored factors affecting adherence [5] and evaluated interventions to increase medication adherence [6,7]. Furniss et al. [8] suggested taking a systems resilience approach to adherence whereby strategies are developed to help the patient remember to take their medicines. Previous research has identified that patients develop their own strategies to assist with adherence [9]. However, such research has not focused on how the helpfulness of these strategies may change in response to changes in patients’ circumstances.

We aimed to explore resilience of medication adherence strategies to life changes, drawing on data obtained from those staying at home as much as possible during the COVID-19 pandemic. 

## 2. Materials and Methods

### 2.1. Data Source

We carried out a secondary analysis of the transcripts of 50 semi-structured interviews that had been carried out in a study investigating medication safety practices for UK patients staying at home as much as possible during the first wave of the COVID-19 pandemic [10]. The study included adults who were advised to “shield” [11] or were over 70 years of age and who were taking at least one long-term medicine. Table 1 describes their characteristics [10]. Any adults that had assisted with the medication management of such adults were also eligible for participation [10]. Those under 18 years, unable to consent to interview, or without access to telephone or internet, were excluded [10]. Participants were recruited through the researchers’ own personal and professional networks, patient and carer charities and organisations, and engagement with patient and public involvement partners [10]. Both social media and word-of-mouth were used to support recruitment. Interviews were conducted by telephone or video conferencing between June and August 2020 and were led by two research pharmacists (SG, BDF) and a health services researcher (CW) who had previous experience conducting qualitative interviews. Participants were asked, using open questions, about their usual home medication practices and how these had changed during the pandemic, including any challenges relating to taking their medicines. Medication adherence was not measured quantitatively.

### 2.2. Data Analysis

Inductive thematic analysis was carried out [12]. The transcripts were read multiple times and data linked to medication adherence practices were highlighted. This process was repeated to ensure minority views were also included. After formation of initial codes, the codes were compared to each other and the original quotes they were generated from. Codes were refined and those with similar concepts grouped together to form potential overarching themes. Connections between the codes were identified and the resulting themes checked to ensure coherence.

One researcher (MA) coded all fifty interviews. A second researcher (RB) independently conducted thematic analysis on, and coded, a random sample of 10% (n = 5) of the interview transcripts. The codes identified by both researchers were compared and were found to be consistent with each other. Neither researcher identified codes that had not been picked up by the other. Both sets of codes were merged and finalised following a discussion between the researchers before the themes were finalised.

### 2.3. Ethical Considerations

NHS ethical approval was not required as participants had not been recruited through the NHS. The study was approved by the UCL ethics committee (ref 18417.001). The participants had all given informed consent. Where possible, this had been through an electronic signature, or a signed hard copy of the consent form being posted. Alternatively, verbal consent had been invited by audio-recorded telephone or video call, as approved by the ethics committee. All the transcripts analysed were anonymised and any identifiable information removed prior to this secondary analysis.

## 3. Results

Participants focused on unintentional nonadherence rather than intentional nonadherence when discussing strategies to overcome challenges to adherence. However, the interview data suggested that there were also some potential or actual changes in intentional adherence.

### 3.1. Resilience of Strategies Used to Reduce Unintentional Nonadherence

Analysis of the interview transcripts led to identification of three types of strategies that patients used to help reduce unintentional nonadherence during normal times: routine, reminders, and multicompartment compliance aids (Figure 1). Routines and multicompartment compliance aids appeared to be more commonly used than reminders. Resilience of these strategies varied once the pandemic began.

#### 3.1.1. Routine

Some participants reported that they had had a daily routine that they kept before the disruption of the pandemic. This enabled them to associate their medication use with their routine, such as before leaving for work, or upon returning home. However, once they needed to stay at home during the pandemic, they could no longer use their regular schedule as a prompt for medication adherence. Participants taking multiple doses of medication a day seemed to be more affected by disruption to routines. For example, taking morning medication later than usual meant that spacing out doses evenly and keeping track of doses became more challenging.


*“Before I did have a sort of routine because I had to take it twice a day. So, I’d like try and wake up in the morning and then I was told to take it like 12 h after so then I’d try to take it just before I went to bed but because of the pandemic I’ve been waking up and going to bed later so I keep forgetting if I’ve taken them, or if I’ve taken them too closely, or have taken them too far apart.”*
(Participant 33).

However, not all those who relied on this medication adherence strategy before the pandemic were affected in this way, particularly when the part of their routine that their medication taking practice was linked to had not changed. For example, patients with diabetes reported that their medication taking was linked to their meals. Despite changes to their normal routine, they ate their meals at the same times of day.


*“Doesn’t affect me at all. I just take my metformin three times a day with my meal.”*
(Participant 7).

Some patients built resilience into medication adherence during the pandemic by linking their medication-taking to a different aspect of their routine that had not been disrupted, although this was not always completely successful. For example, one participant reported using brushing their teeth in the morning as a prompt to take their medicine, after they were no longer going out to work. However, the participant stated that sometimes this did not help them remember whether they have taken their medication or not.


*“I take my tablets after I brush my teeth essentially first thing on a morning So if I haven’t brushed my teeth, I am pretty sure that I haven’t taken my tablets either, sometimes can’t remember whether I have taken my tablets when I have brushed my teeth but that’s another story.”*
(Participant 25).

Some participants reported that during normal times they used visual cues, linked to their routine, where their medication was stored strategically so that it would be visible at the time they needed to take their medication. This would then act as a prompt for medication intake.


*“I know what I’m taking at a certain time of day because they’re close to where I make a cup of tea.”*
(Participant 11).


*“I had to take a statin at night, before I go to sleep, and so actually instead of leaving that box of pills with all the others, I actually put it up in my bedroom and so it was there and I didn’t forget it.”*
(Participant 35).

The type and location of visual cues varied between participants, depending on the time of day their dose was required to be taken and where it was more likely to be noticed. When people started staying at home during the pandemic, visual cues used previously may no longer have been effective, but for some, their system was adaptable and other visual cues were used instead.


*“I guess I store my medications differently now. So, instead of having them all in one space because I would usually try and get up and take the medication and go to bed and take the medication, but because I keep forgetting, I changed the location. So, I’ve got a couple by my bed for when I wake up and then the living room for when I want to go to bed so to try and remember before I go up to bed. I think it kind of helps me remember if I’ve taken it or not. Because they don’t have the days or anything, I get mixed up if I’ve taken them or not and I think because they’re in different locations, I can tell if I’ve taken them-ish.”*
(Participant 33).

#### 3.1.2. Reminders

Some participants reported that they relied on reminders to help adhere to their medication. This could be in the form of a family or carer reminding them, or as an alarm or notification on an electronic device. This system may not have been something participants had been using before or were able to use before, but since they had been staying at home, some were with family members for a much larger part of the day. This meant receiving helpful reminders from family members was possibly more viable as an adherence practice than it was before. In addition, for some participants who had been using this method prior to the pandemic, the effectiveness of this adherence strategy had remained the same during the pandemic.

Conversely, one form of reminder alone was not a strong enough adherence practice for some participants. As well as encouraging family members to remind them, some also attempted to set notifications on their phones.


*“I actually tried to make sure by putting alarm on my phone now.”*
(Participant 22).

However, using alarms or notifications as an adherence practice had limitations for some participants.


*“I probably could set reminders on my phone given that it’s already inundated with the emails and WhatsApps and those sorts of things about Coronavirus, I can’t really have another thing on there as well. There wasn’t the mental space for that.”*
(Participant 3).

#### 3.1.3. Multicompartment Compliance Aids and Calendar Packs

Some participants reported that they used multicompartment compliance aids. This enabled participants to keep track of their medication usage, helped ensured doses were taken at the correct times, and could serve as a reminder of when to reorder.


*“I use a dosette box and otherwise I’m okay with taking my medication.”*
(Participant 50).

The use of multicompartment compliance aids seemed resilient to changes caused by the pandemic, with the majority of those who used them still reporting to find them effective while homebound. When asked whether their medication adherence practice of using multicompartment compliance aids had been affected by the pandemic, the majority reported that it had not. However, one participant had asked a different member of the family to fill the compliance aid [10]. In addition, on a related issue, one participant taking only one medication described how the calendar pack it was usually dispensed in helped him remember whether or not he had taken his medication. He reported that the medication was no longer being supplied in a calendar pack during the pandemic, making this more challenging, with his wife thinking this was probably due to medication shortages.

The use of multicompartment compliance aids was also not a successful adherence practice for everyone. One participant had attempted to use them to support their medication usage but found them ineffective.


*“Like the dosette box, we started to try (to put the tablets) in the compartments, it kind of didn’t work.”*
(Participant 36).

#### 3.1.4. No Specific Strategy

Some participants did not specify any medication adherence practice that was used prior to the pandemic. They reported high adherence levels pre-pandemic, and that they were unaffected by having to remain homebound. Some participants found that their adherence to their medication had improved.


*“No, because I just take them as I normally take them, pandemic or not.”*
(Participant 45).


*“You could say, in fact, that if I didn’t have to go out ever again, I would probably never forget to take my medicine.”*
(Participant 5).

### 3.2. Changes to Intentional Nonadherence during the Pandemic

In addition to the changes associated with unintentional nonadherence, some participants described changes to their beliefs about and motivation to take medication. Two participants discussed a re-evaluation of the risk/benefit ratio of their medication as it was taking the medication that had put them at higher risk from COVID-19 and required them to shield. While neither of these participants reported that they had decided to stop taking their medication at the time of interview, they reported that they were continually evaluating the situation, and the length of time that they were going to be required to shield was reported to be one of the factors that they described as being relevant. In addition, one participant described making a decision to stop one of her medicines because she was no longer experiencing symptoms, and did this without consulting a healthcare professional. She was of the view that she would have consulted a pharmacist during normal times, but she knew that they were extremely busy and therefore found relevant information to support her on the internet. Another participant described feeling more negative about her illness and less motivated during the pandemic, making it more difficult to adhere to her medication. In contrast, other participants reported increasing their adherence during the pandemic to ensure their long-term conditions were fully under control, potentially making them less susceptible to more serious illness.


*“Well, the thought of getting COVID, well, it really concentrated my mind and I made sure I didn’t miss any nebulisers, took my tablets by the clock and I sort, well, if I’m going to get this wretched disease, I want my chest to be in as good a condition as it can be and maybe the medication that I’m taking might actually help.”*
(Participant 43).


*“You start these medications with a good idea of what the risks are and suddenly that’s taken [away] and you’re not sure how to weigh that up anymore. If I’m not supposed to go out ever again until COVID is definitely gone … that means that the risk/benefit balance has completely then fallen apart and then it becomes, is it worth taking them anymore, [if] that’s the level of restriction they can have on my quality of life. So it’s quite a profound, I suppose, effect if you think about it.”*
(Participant 3).

In contrast to unintentional nonadherence, participants did not identify specific strategies that they used to deal with intentional nonadherence. Rather, there was potential for their intentions to change and then affect their medication behaviour.

Figure 2 summarises the themes identified in relation to changes to challenges in adherence experienced during the pandemic.

## 4. Discussion

Resilience of strategies used by participants to increase unintentional nonadherence varied among participants and types of strategy. Where the strategy used was linked to a part of a routine that then altered, it seemed to be less resilient. These practices then had to be adapted to remain effective, or new adherence strategies developed. Some participants were able to make some changes to their strategy to adopt to their new routine and/or make use of other enabling factors, such as spending more time with family who were then able to help them remember to take their medicines. Participants who reported that they did not use any specific strategy to assist with adherence reported that they had not noticed any changes during the pandemic.

These findings build on the approach taken by Furniss et al. [8] in building resilience to nonadherence. Our findings suggest that an extra layer of resilience is needed to ensure that the strategies developed to build resilience are effective long-term, and when life circumstances change. Mikelson and Holden [4] also found that other disruptions to routine, such as being out of the home, could lead to more challenges with adherence. Our findings are supported by those of Musiimenta et al. [13], who found that after withdrawal of text reminders, some participants were unaffected, some adopted a new strategy, and some returned to finding adherence challenging. Our study suggests that this can also be applied to medication adherence strategies that patients put in place themselves. Furthermore, it provides additional information on which strategies are more likely to be susceptible to changes in life circumstances and how these may need to be adapted.

### 4.1. Implications for Practice

Our findings suggest that challenges associated with medication adherence do not always remain stable over time and that healthcare professionals need to continue to monitor and support medication adherence long-term. Healthcare professionals may be aware of major disruption or life changes that may make adherence more challenging, such as an admission to hospital or a bereavement, and thereby provide appropriate support. However, in other cases, healthcare professionals may be unaware of changes, thus regularly reviewing adherence with the patient could be beneficial. In addition, proactively building resilience into the system, even while patients are adherent, may make this susceptible to change as circumstances become more challenging. Perhaps, unsurprisingly, when patients make an intentional decision to not adhere to their medicines, they do not seem to build strategies to overcome this. Proactively discussing patients’ motivation and beliefs about medicines and any changes taking place over time is therefore also important.

### 4.2. Strengths and Limitations

To our knowledge, this is the first study to have explored the effect that major life changes may have on challenges to medication adherence. The sample size of 50 was large for a qualitative study. Nevertheless, some groups were not represented, including patients without access to either the internet or telephone, and those who did not speak English. Another limitation was that adherence was not quantitatively measured.

## 5. Conclusions

Our findings suggest that both intentional and unintentional nonadherence may be affected as a result of changes to life circumstances. Resilience of strategies used by participants to increase unintentional nonadherence in response to such changes vary between participants and type of strategy. In particular, strategies linked to a routine may no longer by effective when that routine changes. Healthcare professionals need to continue to monitor and support medication adherence long-term.

## Figures and Tables

**Figure 1 healthcare-09-01048-f001:**
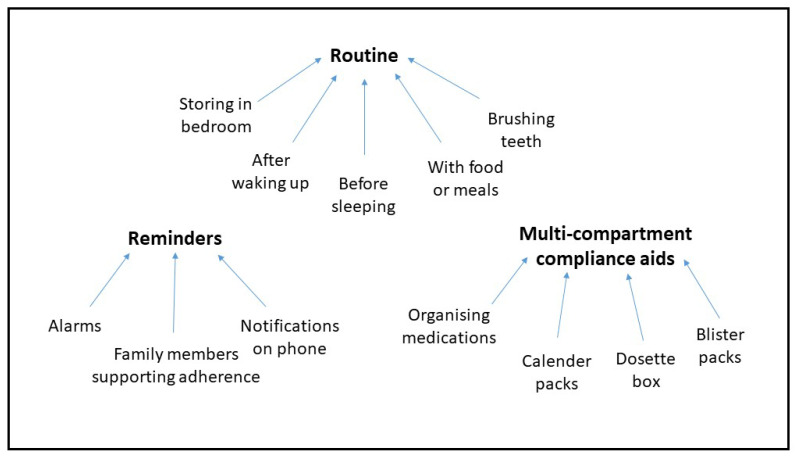
Strategies used by participants to reduce nonadherence.

**Figure 2 healthcare-09-01048-f002:**
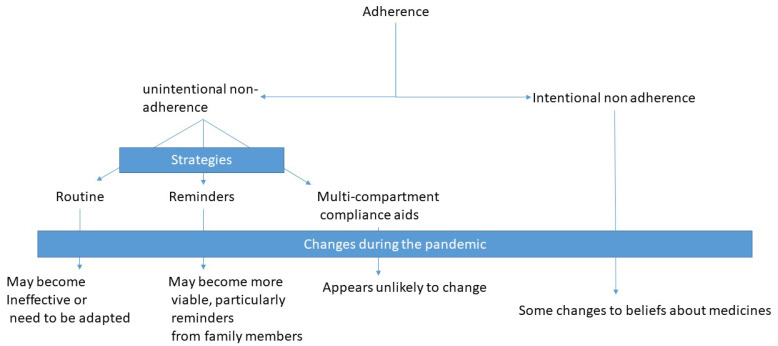
Summary of reported changes to challenges experienced with medication adherence.

**Table 1 healthcare-09-01048-t001:** Participants’ characteristics.

Characteristic	Distribution
Gender	16 males 34 females
Age	Mean age: 68 (range 26–93)
Ethnic group	43 white 7 other ethnicities
Urban versus rural location	43 urban 1 semi-rural 6 rural
Number of medicines	Range 1–17
Living arrangements	40 living with others 10 living alone
Dominant role in assisting other family member with medicines	9 yes 41 no

## Data Availability

Participants did not consent to having their full transcripts publicly available. However, anonymised excerpts are available from the authors who may be contacted on s.garfield@ucl.ac.uk.

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
