# Peer review of "Resilience of Medication Adherence Practices in Response to Life Changes: Learning from Qualitative Data Obtained during the COVID-19 Pandemic"

_healthcare, 2021, doi:10.3390/healthcare9081048_

Round 1

Reviewer 1 Report

Thank you very much for giving an opportunity to review the present manuscript. Authors aimed to explore resilience of medication adherence strategies to life changes during COVID-19 pandemic, and found that both intentional and unintentional non-adherence may be affected by result of changes to life circumstances. Since no studies have assessed how the helpfulness of these strategies may change in response to changes in patients’ circumstances, their view point must be important.

There are some improvements that should be made before publication.

  1. Authors should present 50 participants background data in addition to age and sex (e.g. underlying disease, types of medicine, and number of medicines). If possible, it will be easier to understand if it is shown in a table.
  2. How did the authors assess medication adherence and its changes?

Author Response

Thank you very much for giving an opportunity to review the present manuscript. Authors aimed to explore resilience of medication adherence strategies to life changes during COVID-19 pandemic, and found that both intentional and unintentional non-adherence may be affected by result of changes to life circumstances. Since no studies have assessed how the helpfulness of these strategies may change in response to changes in patients’ circumstances, their view point must be important.

We thank the reviewer for their positive comments and are pleased that they found our paper to be important.

Authors should present 50 participants background data in addition to age and sex (e.g. underlying disease, types of medicine, and number of medicines). If possible, it will be easier to understand if it is shown in a table.

Thank-you for these suggestions.  We have now added a table including more background data, including number of medicines.  We did not collect data on disease type or type of medicines.

How did the authors assess medication adherence and its changes?

We explored challenges to medication adherence qualitatively through semi-structured interviews.  We have now clarified this in the paper and explicitly stated that medication adherence was not assessed quantitatively.

Reviewer 2 Report

In this manuscript the authors conducted a secondary analysis of qualitative interview data from 50 participants to investigate the strategies the participants used for medication adherence during COVID-19 pandemic.  Overall, this manuscript dealt with an important topic and contributed to our understanding of medication adherence.  Having said that, there are some issues to be addressed by the authors.

(1) It will be helpful if authors can provide a figure of the coding scheme of the qualitative data, especially connections between the codes.

(2) While the authors were conducting a secondary analysis of qualitative interview data, it will be helpful if the authors can provide exact number of participants (out of 50) instead of using “many participants” or “some participants” in many places in the manuscript.

(3) The authors should revise the sentence in lines 18-21 in the abstract.  I don’t feel comfortable when reading it.

Author Response

Overall, this manuscript dealt with an important topic and contributed to our understanding of medication adherence.

We thank the reviewer for their positive comments and are pleased that they found our paper to be important.

It will be helpful if authors can provide a figure of the coding scheme of the qualitative data, especially connections between the codes.

We have now provided 2 figures representing the coding system, including connections between codes.  We feel that these enhance the paper and thank-you the reviewer for this suggestion.

While the authors were conducting a secondary analysis of qualitative interview data, it will be helpful if the authors can provide exact number of participants (out of 50) instead of using “many participants” or “some participants” in many places in the manuscript.

We have given full consideration to this suggestion.   However, as interviews were semi-structured and we employed thematic rather than content analysis, we do not feel that adding exact numbers would be meaningful or appropriate in relation to such qualitative analysis. However, we had already given an indication of which viewpoints were reported by fewer respondents and have now clarified this further.

The authors should revise the sentence in lines 18-21 in the abstract.  I don’t feel comfortable when reading it.

We have revised this sentence to make it clearer.

Round 2

Reviewer 2 Report

In the revised manuscript the authors fully addressed my first comment. 

For the second comment, while the authors gave reasons why they would not provide exact number of participants in each scenario, I still expect the authors to clarify what each descriptive word means.  For example, I suppose “most” means over 90% cases, “many” means a majority, i.e., over 50% but less than 90% cases, “some” means a minority, i.e., less than 50% but over 10% cases, and “few” means less than 10% cases.  When I use NVivo to conduct thematic analysis, it still tells me how many times each code is found in all cases.

For the third comment, I am not comfortable with clauses starting with “some that…”.
